# Machine Learning-Assisted Classification of Paraffin-Embedded Brain Tumors with Raman Spectroscopy

**DOI:** 10.3390/brainsci14040301

**Published:** 2024-03-23

**Authors:** Gilbert Georg Klamminger, Laurent Mombaerts, Françoise Kemp, Finn Jelke, Karoline Klein, Rédouane Slimani, Giulia Mirizzi, Andreas Husch, Frank Hertel, Michel Mittelbronn, Felix B. Kleine Borgmann

**Affiliations:** 1Department of General and Special Pathology, Saarland University (USAAR), 66424 Homburg, Germany; 2Department of General and Special Pathology, Saarland University Medical Center (UKS), 66424 Homburg, Germany; 3National Center of Pathology (NCP), Laboratoire National de Santé (LNS), 3555 Dudelange, Luxembourg; 4Luxembourg Center of Neuropathology (LCNP), 3555 Dudelange, Luxembourg; 5National Center of Neurosurgery, Centre Hospitalier de Luxembourg (CHL), 1210 Luxembourg, Luxembourg; 6Doctoral School in Science and Engineering (DSSE), University of Luxembourg (UL), 4362 Esch-sur-Alzette, Luxembourg; 7Faculty of Medicine, Saarland University (USAAR), 66424 Homburg, Germany; 8Department of Cancer Research (DoCR), Luxembourg Institute of Health (LIH), 1210 Luxembourg, Luxembourg; 9Luxembourg Centre for Systems Biomedicine (LCSB), University of Luxembourg (UL), 4362 Esch-sur-Alzette, Luxembourg; 10Department of Life Sciences and Medicine (DLSM), University of Luxembourg, 4365 Esch-sur-Alzette, Luxembourg; 11Faculty of Science, Technology and Medicine (FSTM), University of Luxembourg, 4365 Esch-sur-Alzette, Luxembourg; 12Hôpitaux Robert Schuman, 1130 Luxembourg, Luxembourg

**Keywords:** Raman spectroscopy, vibrational spectroscopy, gliomas, meningiomas, brain metastasis, tumor necrosis, artificial intelligence, machine learning, random forest classification

## Abstract

Raman spectroscopy (RS) has demonstrated its utility in neurooncological diagnostics, spanning from intraoperative tumor detection to the analysis of tissue samples peri- and postoperatively. In this study, we employed Raman spectroscopy (RS) to monitor alterations in the molecular vibrational characteristics of a broad range of formalin-fixed, paraffin-embedded (FFPE) intracranial neoplasms (including primary brain tumors and meningiomas, as well as brain metastases) and considered specific challenges when employing RS on FFPE tissue during the routine neuropathological workflow. We spectroscopically measured 82 intracranial neoplasms on CaF_2_ slides (in total, 679 individual measurements) and set up a machine learning framework to classify spectral characteristics by splitting our data into training cohorts and external validation cohorts. The effectiveness of our machine learning algorithms was assessed by using common performance metrics such as AUROC and AUPR values. With our trained random forest algorithms, we distinguished among various types of gliomas and identified the primary origin in cases of brain metastases. Moreover, we spectroscopically diagnosed tumor types by using biopsy fragments of pure necrotic tissue, a task unattainable through conventional light microscopy. In order to address misclassifications and enhance the assessment of our models, we sought out significant Raman bands suitable for tumor identification. Through the validation phase, we affirmed a considerable complexity within the spectroscopic data, potentially arising not only from the biological tissue subjected to a rigorous chemical procedure but also from residual components of the fixation and paraffin-embedding process. The present study demonstrates not only the potential applications but also the constraints of RS as a diagnostic tool in neuropathology, considering the challenges associated with conducting vibrational spectroscopic analysis on formalin-fixed, paraffin-embedded (FFPE) tissue.

## 1. Introduction

Approaches to more precise and observer-independent tissue diagnostics have been largely developed in the last decades—evolving from the first use of immunohistochemistry (1942) to the implementation of genetic parameters in tumor diagnostic to methylome-based tumor class prediction in the current WHO brain tumor classification [1,2]. To date, histopathological tissue analysis in combination with epigenetic tumor classification is the defined “gold-standard” of brain tumor diagnosis, enabling not only solid consecutive treatment decisions but also evidence-based prognoses. As an alternative attempt for reliable tumor diagnostics, the vibrational spectroscopic technique named Raman spectroscopy (RS) has gained increased research interest in the neurooncological field. This optical method detects changes in the molecular vibrational state of tissue samples and thus uses spectral tissue properties to, e.g., determine tumor origin or to gain insights into tumor biology [3]. So far, the practical potential of this method in combination with supervised and unsupervised machine learning algorithms has been shown not only for the classification of tumor types and distinct histomorphological tumor areas but also for the differentiation of the tumor grade or the detection of tumor margins/infiltration zone [4,5,6,7]. Most recently, Zhang et al. yielded >80% accuracy with support vector machine-based intraoperative differentiation of glioma tissue and healthy control, and Romanishkin et al. reported a support vector machine-based accuracy in detecting glioblastoma tissue (regardless of the histologically proven sampling area, namely, central tumor core or tumor edge) of 83% [8,9]; by using unprocessed samples of pediatric brain tumors, even an accuracy of 86.2% was achieved when classifying between low-grade gliomas and normal brain by employing a logistic regression model [10]. Although, commonly, fresh tissue specimens are spectroscopically examined intraoperatively in vivo [11] by using a handheld Raman probe or ex vivo [4,12] by using advanced Raman imaging techniques, e.g., Stimulated Raman Histology, other approaches aim to establish RS on formalin-fixed or formalin-fixed and paraffin-embedded (FFPE) tissue. The latter is accompanied by challenges such as degradation/fragmentation of nucleic acids and protein cross-linking within FFPE tissue [13,14,15,16], strong spectral signal of paraffin wax adulterating the biological Raman bands [17], modification of biological lipids related to the dewaxing process [18], and occurrence of fluorescence due to silicon-based glass slides [13], which should be addressed pre-experimentally and also considered during data processing and analysis. Nevertheless, RS has been successfully applied on FFPE tissue to classify distinct tumor types [19,20], to predict the genetic status of IDH-mutant/wildtype astrocytomas [21], and to distinguish certain histological areas in the morphological heterogenous glioblastoma based on individual spectral properties [22]. In order to further exploit the potential of RS as an additional method in neuropathology, we acquired the Raman spectra of distinct histomorphological areas (vital tumor and necrosis) of a broad range of intracranial neoplasms and classified them using an in-house-built machine learning pipeline. Within our approach, we did not solely aim at the discrimination of different tumor groups; we further evaluated the question of whether distinct histomorphological areas remain detectable due to their spectroscopic behavior even when compared with areas of another tumor type.

## 2. Materials and Methods

### 2.1. Patient Data

Sample specimens of 82 primary brain tumor cases (astrocytoma IDH mutant, oligodendroglioma 1p/19q co-deleted, ependymoma, glioblastoma IDH wildtype; see Appendix A for clinical data), meningeal tumors (transitional/meningothelial meningiomas), and brain metastases of different primary origin (breast, colorectal, and lung (non-small cell lung cancer (NSCLC)) carcinomas) were examined and measured by means of RS. See Table 1 for an overview of the included tumor groups/types and the corresponding Raman measurements carried out. The samples included are part of the INSITU^®^ study (approved by the “Comité National d’Ethique de Recherche’”, No. 201804/08) in Luxembourg, and all patients involved gave their written formal consent for tissue usage. The spectroscopic examination of tissue samples was exclusively carried out at Laboratoire national de santé (LNS; Luxembourg) after routine neuropathological diagnostic workup (light microscopy, IHC, and genetic and epigenetic testing) by a board-certified neuropathologist (M.M.). 

### 2.2. Sample Preparation

FFPE tissue blocks were selected according to the neuropathological diagnosis and histomorphological description; thereupon, the tissue blocks were sequentially cut two times (each section of 7 μm), using a standard microtome. While the first section was stained (H&E—hematoxylin and eosin) and thus amenable to light microscopic examination, the second section was not stained to avoid additional negative impact on the biological spectral signal. Instead of using a silicon-based glass slide, which would have affected the Raman measurements due to its strong fluorescence background, the second section was placed on a CaF_2_ slide (calcium fluoride; Crystran, Poole, UK) [13]. With its beneficial property of only one single Raman band at 321 cm^−1^, the CaF_2_ slide allows for the spectroscopic assessment of biological tissue [23,24]. In a next step, the tumor sample of the second section was subjected to a deparaffinization process, in which residual paraffin wax was physically (by briefly heating it) and chemically (by means of repetitive xylene baths) reduced. This step is crucial due to the strong Raman signal (distinct bands at 1063, 1133, 1296, and 1441 cm^−1^) of paraffin wax, which would potentially overlap with the genuine Raman spectrum of the tumor sample [17,25]. See Appendix A for the detailed deparaffination protocol. Finally, the H&E-stained section was light-microscopically analyzed, and the histomorphological areas of interest (vital tissue and necrosis) were marked with a fine marker pen. This marking on the H&E slide allows for an accessible transfer of the localization of the desired tumor area to the subsequent unstained slide, where the corresponding area could also be marked and afterwards submitted to spectroscopic examination. Post-experimentally, the reusable CaF_2_ slides were hand-washed with a rinsing agent (containing, among others, surface-active agents and ethanol) and warm water until remaining tumor tissue was no longer visible; this was performed in between measurements for quality assurance by means of spectroscopy exclusion. 

In this study, two distinct histomorphological areas of interest were defined. The first area was a vital tumor zone, while the second one consisted only of necroses. Due to its occurrence in only a few tumor types, it was spectroscopically assessed only whenever feasible (i.e., glioblastoma and brain metastases). The area of necrosis was defined as an eosinophilic area lacking vital tumor cell nuclei. 

### 2.3. Data Acquisition and Raman Spectroscopy

After pre-experimental preparations were completed, the CaF_2_ slides with the unstained tissue were submitted to spectroscopic measurements. For the accurate selection of the measurement point, the laser spot of the Raman spectrometer was placed within the marked area in a visually controlled manner. The actual RS measurement was performed in an in-house-built dark chamber with the aim to reduce external sources of light interference, thereby preserving the biological RS signal [26,27]. With a laser spot size of 100 μm, a lens including a working distance of 7 mm and the chosen acquisition parameters of 10 s, 30 averages, and 90 mW (laser power) per measurement, we were able to achieve a fast readout. Therefore, in our approach, a marked area was measured repeatedly, but at different spots within one area. All Raman measurements were conducted using the portable ProRaman-L high-performance Raman spectrometer (TSI, Shoreview, MN, USA) including a CCD detector system; see Appendix A. For data acquisition, the default ProRaman Reader software, version 8.3.6 (TSI, Shoreview, MN), was employed, and data smoothing was applied directly after each measurement by using a default Savitzky–Golay filter. All data were stored as .spc files and made accessible to data analysis in a next step. See Figure 1 for a general overview of our study protocol.

### 2.4. Machine Learning 

Starting with data preprocessing, all measurements were visualized (Appendix A displays all raw Raman Spectra sorted by tumor type) and internally checked for outliers due to non-biological confounding factors, e.g., cosmic ray artifacts and hot pixels [28]. In a next step, input data were standardized and trends removed using a Savitzky–Golay filter; afterwards, the Raman spectra were cropped to the range between 500 cm^−1^ and 2000 cm^−1^. This spectral range, also known as the “biological area”, is widely used in spectroscopic studies due to a great contribution of Raman bands resulting from underlying biological tissue. See Figure 2 for mean Raman spectra of primary brain tumors, meningiomas, and brain metastases. For deeper machine learning-based analysis, we used an internally written Python script in which we set up a machine learning pipeline to enable iterative classification with divergent input data based on each emerging classification task. In this manner, different research questions can be approached with consistent, a priori-selected parameters and types of validation. To avoid patient-dependent clustering, the input data were split in a training cohort (75% of data) and an external validation cohort (25% of data) containing only Raman measurements of different patients. In the course of this, patient identity did not serve as prediction output but rather for confounder variable control; within the scope of the study, strictly the identity of the area measured by one single measurement was determined as output. After the training of random forest (RF) classification, containing internal random cross-validation, as the selected model within our pipeline, the resulting algorithm’s performance was assessed. To do so, the accuracy, the ROC (receiver operating characteristic) curve, the corresponding AUROC (area under receiver operating curve) value, and the PR (precision–recall) curve, including the corresponding AUPR value, were plotted for each classification decision threshold; furthermore, the misclassification ratio (misclassified counts relative to total measurements per class) and the scores of the commonly used metrics precision (positive predictive value), recall (sensitivity), and F1-score (mean of precision and recall) were reported. To gain a deeper insight into model performance, the macro average (average based on all individual classes equally) and the weighted average (average considering the quantities of measurements in each class) were calculated for each of the three metrics. See Appendix A for a more detailed description of our machine learning workflow.

## 3. Results

### 3.1. Multi-Class Classification for Discrimination of Tumor Origin

The optimized RF model was first trained to address an initial multi-class classification task. The pathologically determined origin of each tumor served as class and thus as histological ground truth for the model; in our case, this resulted in the following three classes: primary brain tumors (here: gliomas), carcinoma brain metastases, and meningeal tumors (here: meningiomas). At this point, only RS measurements of vital tumor tissue were considered. The splitting numbers for the training and the external validation cohorts, as well as the total number of patients and measurements, are displayed in Appendix A. The trained model was capable of detecting differences in the spectral properties of tumors of varying origin with a macro-average AUROC value of 0.76 (AUROC value for each class: glioma, 0.84; carcinoma brain metastases, 0.76; meningioma, 0.66) and an overall accuracy of 63%. See Figure 3A for the resulting ROC and PR curves, as well as the AUROC and AUPR values. A confusion matrix with predicted observations in the test set and the corresponding metrics of precision, recall, and F1-score are shown in Figure 3B,C. The primary brain tumor group and the carcinoma brain metastasis group can be clearly distinguished as individual classes based on their spectral properties (misclassification ratios of 0.11 and 0.52, respectively), while the meningioma class was frequently misclassified, with a ratio 0.85. To investigate this distinction in more detail, we repeated the model training process with the same measurements of primary brain tumors and carcinoma brain metastases as input data; this time, each tumor subclass (astrocytoma, oligodendroglioma, ependymoma, glioblastoma, NSCLC metastasis, colorectal carcinoma metastasis, and breast carcinoma metastasis) served as the histological ground truth and thus as class (output) in our learning model. Even in this multi-class (seven classes) learning approach, the primary side of primary brain tumors and carcinoma brain metastases is still reliably detectable, since misclassifications between the tumor types occur most likely within their biological tumor origin (Figure 3D and Appendix A).

### 3.2. A Practical Approach: Carcinoma Metastases and Glioma Classifier

Upon microscopical identification of a brain tumor as likely metastatic in origin, the question of the primary origin arises. This question is not easily addressed by means of basic light microscopy and frequently requires immunohistochemistry and further molecular pathology. Similarly, in case of a glial tumor, the identification of different tumor types is sometimes difficult by means of light microscopy alone. Fast and label-free identification based on spectral properties could be very helpful. We, therefore, used all the acquired measurements of vital tumor areas of carcinoma brain metastases (Appendix A) to establish a classifier within our machine learning pipeline to determine the primary tumor origin (lung/breast/colorectal) only by means of spectroscopy. Our trained RF algorithm was able to distinguish between different carcinoma metastases with a macro-average AUROC value of 0.87 (AUROC value for each class: breast carcinoma metastases, 1.00; colorectal cancer metastases, 0.80; and NSCLC, 0.76) and an overall accuracy of 68%, predicting the class of brain metastases of breast tumors with highest scores (precision 1.00, recall 0.83, misclassification ratio 0.17); see Figure 4A for the corresponding ROC/PR curve and confusion matrix. Focusing on the detection of the glioma tumor type and thus also on tumor genetics (IDH mutation in the case of IDH mutant astrocytoma, lack of IDH mutation in the case of IDH wildtype glioblastoma, and 1p/19q co-deletion in the case of oligodendroglioma [29]; see Table 1 and Appendix A), we applied RS to different grades and types of glioma specimens (input data; Appendix A) and were able to predict IDH mutant astrocytomas (AUROC of 0.92, recall of 0.72, precision of 0.57, and misclassification ratio of 0.28) and 1p/19q co-deleted oligodendrogliomas (AUROC of 0.86, recall of 0.50, precision of 0.58, and misclassification ratio of 0.50) in a multi-class classification approach. IDH wildtype glioblastoma and ependymomas were detected with AUROC values of 0.79 and 0.91, respectively; the misclassification ratio was 0.81 for the glioblastoma class and 0.00 for the ependymoma class; see Figure 4B and Appendix A.

### 3.3. A Practical Approach: Classification of Tumor Necrosis

In case the resected tumor samples available for pathological analysis turn out to be purely necrotic during light microscopical examination (e.g., biopsy from central region of a glioblastoma), repeated resectioning or biopsy is necessary for adequate pathological diagnostics. Hence, we evaluated the question of whether RS would be suitable for determining the underlying tumor origin in case of purely necrotic tissue; therefore, we spectroscopically examined areas of necrosis in glioblastoma and in carcinoma brain metastases. By using the established computational learning pipeline, a binary RF classification was able to determine the tumor origin solely based on the spectroscopic properties of fragments of tumor necrosis with an overall accuracy of 92% (macro-average AUROC vale of 0.98) at the best classification threshold. Figure 5 provides a visualization of this practical classification approach and shows the performance evaluation in more detail (see additionally Appendix A). Since both tumor groups have shown distinct spectral properties usable for tumor detection on vital tissue in recent analyses (see above), we investigated whether not only the tumor origin (glioma/brain metastasis) but also the histomorphological appearance (vital tumor/necrosis) of each tumor type remains predictable due to an inherent spectroscopic behavior. Indeed, we could show that all four classes (vital brain metastases, necrosis of brain metastases, vital glioma, and necrosis of glioblastoma) were separable based only on their spectroscopic features (macro-average AUROC vale of 89%; see especially Appendix A for more detailed performance metrics). For a general data overview, refer to Appendix A.

In summary, our results prove the general feasibility of utilizing FFPE tumor samples for vibrational spectroscopic examinations. Moreover, we show that rapid spectroscopic analysis and subsequent identification of the biological origin of tumors remain feasible though dependent on the specific classification task.

## 4. Discussion

In this study, we demonstrated that the employment of RS on FFPE tissue in conjunction with machine learning algorithms allows for a rough discrimination of tumor origin and tumor types depending on the use case. We showed that spectroscopic classification proved feasible even based on necrotic tissue fragments alone. A further development of this technique could prove highly practical in those cases where advanced and time-consuming methods of histopathology would need to be employed and possibilities of initial H&E staining are already exploited. Notably, the fact that RS can be used on purely necrotic samples opens avenues of diagnostics where molecular approaches fail. In our study, necrosis stood out as a separate class both between and within tumor classes; it was possible to distinguish both the neoplastic origin of necrotic tissue and the difference between necrotic and vital tissue within one tumor class. Our findings are in line with previous studies examining tumor necroses by means of optical spectroscopy; already in 2007, Amharref et al. described the presence of spectroscopic features of necrotic tumor areas [30]. Subsequently, Kalkanis et al. determined the good separability of normal brain tissue, glioblastoma, and necrosis with an overall accuracy of 97.8% (accuracy of 77.5% in the case of artifact inclusion) based on a supervised discriminant function analysis in frozen tissue sections; Kast et al. established Raman-based images of different brain regions (including tumor center and necrosis), and our group recently described the solid separability of morphological glioblastoma areas (peritumoral area, tumor core, and necrosis) in FFPE tissue samples with an overall accuracy of 70.5% employing support vector machine based-classification [7,22,31]. Additional analyses are required to prove if our attempt also holds true for the detection of necrosis in a multi-class classification approach containing various types of necrosis (e.g., necrosis of primary CNS lymphoma, necrosis post radiation treatment, and necrosis in non-cancerous diseases). Prospectively, the spectroscopic behavior of peritumoral infiltration zones and non-tumorous brain tissue, which is not greatly available since brain tumor surgery aims for maximal tumor resection with corresponding minimal loss of healthy brain function, also needs to be profoundly integrated in such an approach. On that note, a study conducted by the team of Jermyn et al. already evaluated the spectroscopic properties of unprocessed peritumoral brain tissue and the minimal resolution of vibrational spectroscopy for tumor cell identification, showing an overall accuracy of 92% in detecting glioma cells spectroscopically, which resulted in the spectroscopic recognition of tumor cell margins up to ~3.7 cm or ~2.4 cm (depending on the MRI sequence) more precise than MRI T1- and T2-weighted tumor margin detection, respectively [32]. 

Our results demonstrate that RS can potentially serve as a label-free and easy-to-apply technique for the determination of the tumor type or primary tumor origin even when using FFPE tissue; this fast approach for tissue assessment can be applied without awaiting further molecular analysis. Our presented performance metrics are in alignment with the limited evidence known so far. Fullwood et al. published a principal component-based linear discriminant analysis when applying RS on FFPE tissue and differentiated among different primary sites of brain tumor metastasis (breast/colon/melanoma/lung/esophagus and stomach) with 63% correctly classified spectra overall and 72% correctly classified spectra when differentiating among glioblastoma, brain metastasis, and normal brain tissue; the latter could even be improved when averaging spectroscopic measurements [20]. Within our classification models, an internal classification tendency towards the glioma group can be detected, and a more accurate classification of IDH mutant astrocytomas and ependymomas than glioblastoma can be determined when solely classifying different types of gliomas. The latter is in line with the heterogenous nature of glioblastoma, which makes classification as an individual class difficult, as we previously described in unprocessed glioblastoma specimens as well as FFPE tissue [22,33]. In comparison to our reported glioma classifier aiming at differentiating among a broad range of primary brain tumors, Quesnel et al. used RS and support vector machine-based analysis to differentiate among different grades of gliomas and between tumors with the IDH1 mutation or IDH wildtype, with reported accuracies between 75% and 85% [34].

Since we deal with strongly chemically affected and degraded FFPE tissue, several circumstances and limitations need to be considered when evaluating classification performance and addressing occurring misclassifications. 

One possibility is that the subsequent loss of biological substance classes useful for tumor determination in native tissue [5], such as lipids and small molecules, reduces tumor-specific spectroscopic properties and accordingly limits accurate class assignment. Additionally, residues of paraffin wax, as well as tissue effects related to the fixation process, may potentially affect each spectrum in a uniform manner and consecutively hamper spectral individuality. Besides promising results for the detection of gliomas and brain metastases, the group of meningeal tumors (meningothelial and transitional meningiomas) does not allow for comparable spectroscopic tissue identification in our study. Nevertheless, other studies on native/frozen tissue samples do describe spectroscopic properties of a meningioma tumor class not only in comparison to dura mater [35,36] but also among different grades [37,38] or in comparison to smear diagnostics [26]. Within our initial three-class classification approach, the inherent susceptibility of the collagen-rich matrix of meningiomas to technical artifacts arising, for example, from the fixation/deparaffination process may have contributed to the reported misclassification rate. Besides classification, RS can provide valuable insights into tumor biology. The spectroscopic elements (i.e., Raman bands at a distinct wavenumber) representing important features of a classification model can hint at a molecule of origin. As an example, the spectral footprint of the protein/lipid ratio in glioma was used as a predictor of malignancy [5]. We analyzed our trained models for feature importance and found that our models used a wide range of features throughout the whole spectral range (see Appendix A for the listed Raman bands and corresponding importance for classification), indicating high complexity within the acquired Raman data. Due to the use of chemically treated tissue samples and the spectral complexity observed, we refrain from interpreting Raman bands and defining a spectrum as a diagnostic “fingerprint” for classification (diagnostic approach). In order to further analyze the underlying biochemistry of a sample (exploratory approach), the employment of RS on native, unprocessed, or cryofixed tissue proves favorable due to absence of preceding sample modification [13,39]. 

## 5. Conclusions

In this study, we did not only aim to assess the application of RS during the routine neuropathological workflow but also determined distinct special use cases in which RS may potentially play an advantageous future role in pathological diagnostics. Since all measurements were conducted on biologically highly altered FFPE tumor tissue, our results indicate, as a proof of concept, not only that FFPE tissue samples are generally suitable for vibrational spectroscopic examinations but also that rapid spectroscopical examination and successive determination of biological tumor origin is still feasible—though in accordance with the classification task. A fast insight into tumor origin directly after H&E microscopy potentially helps to communicate preliminary findings to the treating oncologist, focusing the search for primary tumor origin more efficiently without having to wait for more advanced and precise techniques. Using RS, this can be done even on fully necrotic resection material that is otherwise not usable for molecular and histopathological approaches. 

## Figures and Tables

**Figure 1 brainsci-14-00301-f001:**
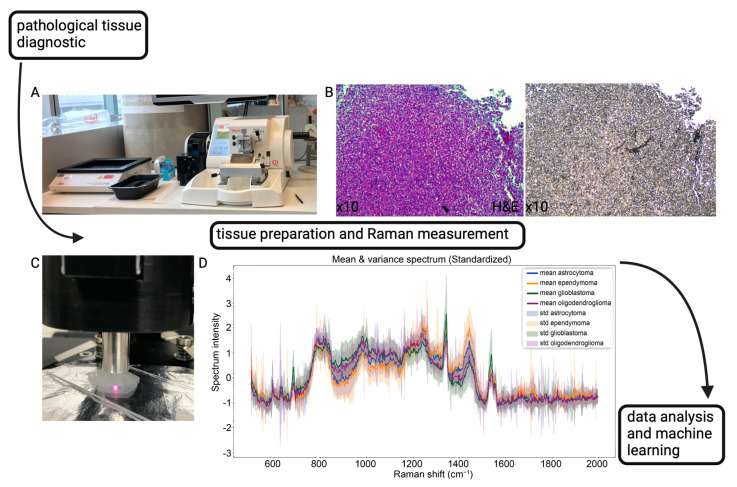
General overview of our experimental workflow. (**A**) Standard microtome used for cutting FFPE blocks. (**B**) Left: H&E-stained glioblastoma tissue on a glass slide. Right: consecutive section (unstained) on a CaF2 slide. (**C**) Raman measurement. (**D**) Mean Raman spectra of various primary brain tumors.

**Figure 2 brainsci-14-00301-f002:**
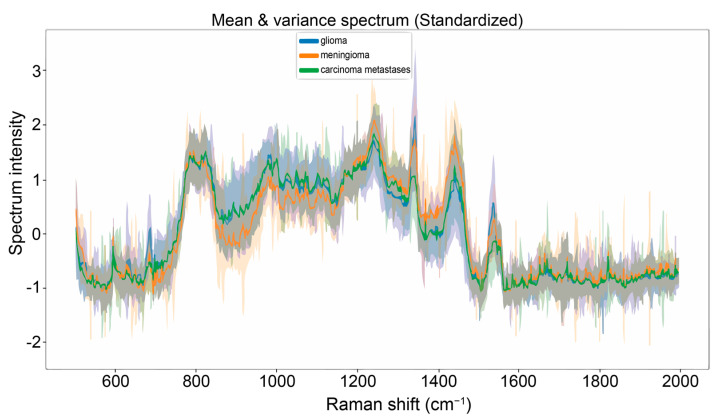
Mean Raman spectra (range: 500 cm^−1^–2000 cm^−1^) for gliomas (blue), meningiomas (orange), and carcinoma brain metastases (green).

**Figure 3 brainsci-14-00301-f003:**
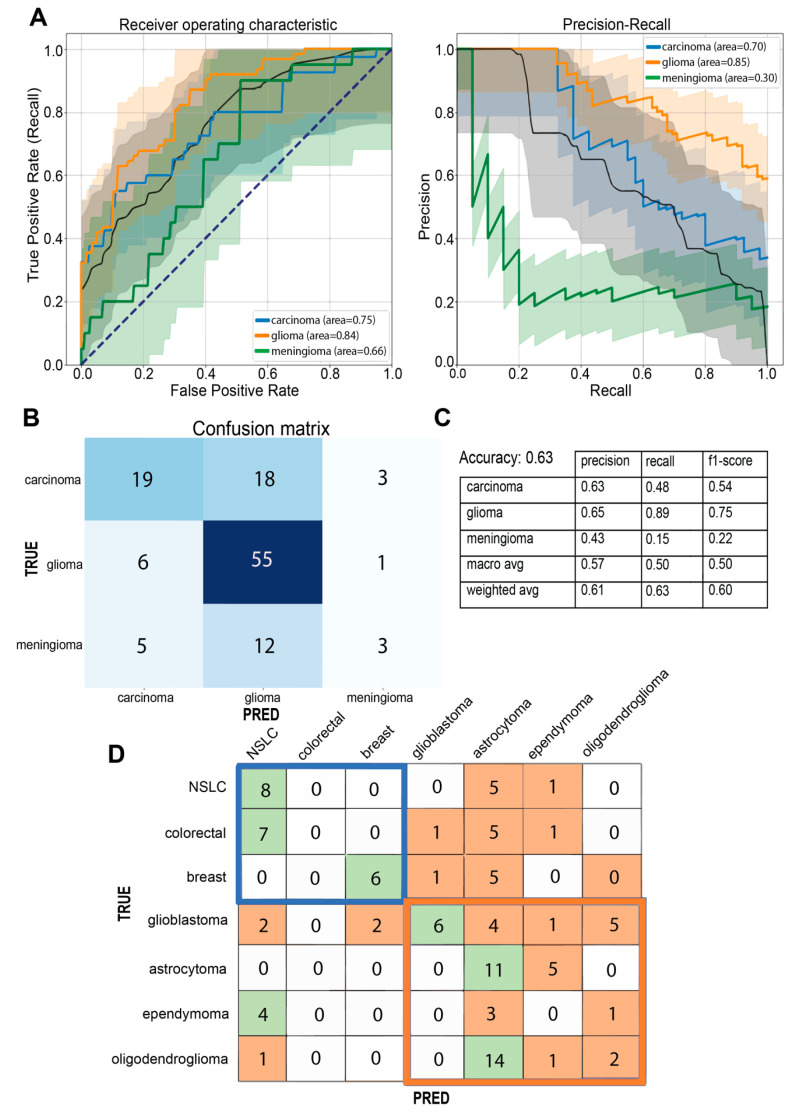
ROC and PR curves of the initial 3-class classification task (black dotted line = random performance level) (**A**) with their corresponding confusion matrix, (**B**) as well as metrics of precision, recall, and F1-score (**C**). Confusion matrix of the second multi-class classification (**D**). Within the 7-class classification model, a preserved assignment to the primary tumor origin (blue box = tumor origin: carcinoma brain metastases; orange box = tumor origin: primary brain tumors) can be seen, despite a notable confusion among the individual tumor types. Green shading = highest-class prediction; orange shading = lower-class prediction.

**Figure 4 brainsci-14-00301-f004:**
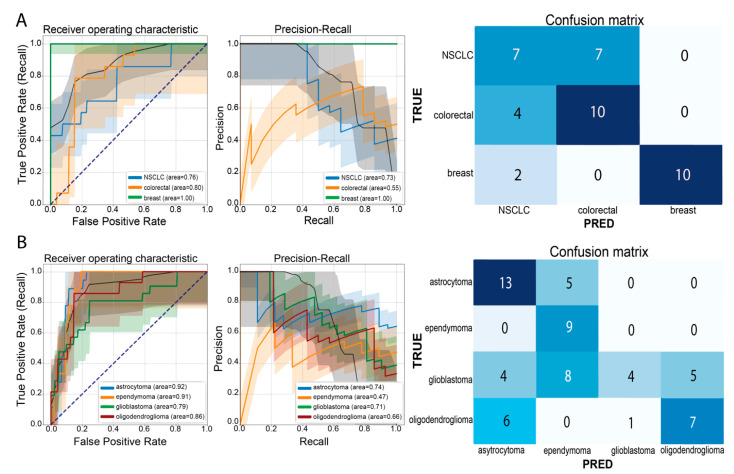
Above: (**A**) ROC/PR curves and corresponding AUROC/AUPR values for the carcinoma brain metastases classifier, as well as the corresponding confusion matrix. Bottom: (**B**) ROC/PR curves and corresponding AUROC/AUPR values for the glioma classifier with the corresponding confusion matrix. Black dotted line = random performance level.

**Figure 5 brainsci-14-00301-f005:**
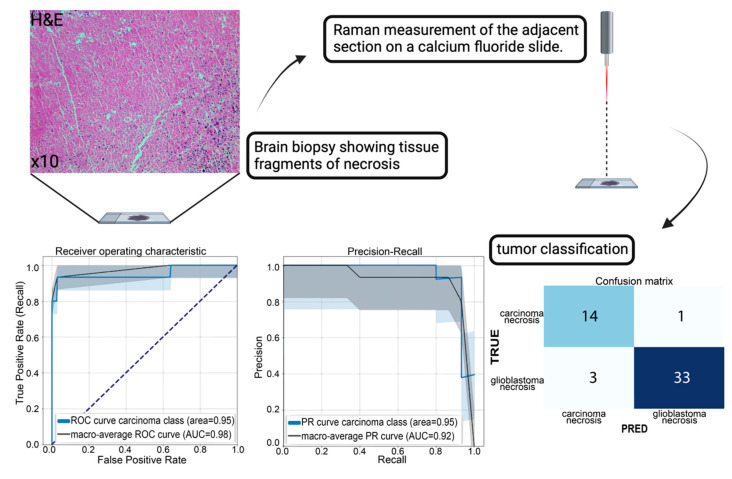
Visualization of the RS approach and subsequent data analyses. In case of only necrotic tissue fragments, an RS measurement followed by machine learning classification may determine the tumor origin. Left upper corner: H&E image of necrosis in a glioblastoma; right bottom corner: ROC/PR curves and confusion matrix of our established binary necrosis classification, showing the good separability based on spectral properties—the different color lines express the individual class performances in contrast to the random performance level (black dotted line).

**Table 1 brainsci-14-00301-t001:** Overview of different tumor types measured by means of Raman spectroscopy.

Tumor Group/ Tumor Type	Number of Cases *n* = 82	Number of Measurements *n* = 679
Astrocytoma of grades 2,3, IDH mutant	9	74
Oligodendroglioma of grades 2,3, 1p19q co-deleted	7	60
Ependymoma	5	44
Glioblastoma, IDH wildtype	27	179
Meningothelial meningioma	4	36
Transitional meningioma	6	56
Breast carcinoma metastases	8	53
Colorectal carcinoma metastases	6	65
Non-small cell lung carcinoma (NSCLC) metastases	10	112

## Data Availability

The data presented in this study are available upon request from the corresponding authors. The data are not publicly available due to the ethical approval, which does not permit the public sharing of data.

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
