# Peer review of "Machine Learning-Assisted Classification of Paraffin-Embedded Brain Tumors with Raman Spectroscopy"

_brainsci, 2024, doi:10.3390/brainsci14040301_

Round 1

Reviewer 1 Report

Comments and Suggestions for Authors

Machine-Learning Assisted Classification of Paraffin-Embedded Brain Tumors Using Raman Spectroscopy
* For some reason, the paper's title in the PDF file does not match the one in the reviewing system. I assume that the authors might have updated their submission at some point and that the editors did not mind that.

The considered manuscript assesses the application of Raman Spectroscopy (RS) to formalin fixed paraffin-embedded (FFPE) intracranial neoplasms for classification of tumors into 7 classes. In this, the authors rely on the self-collected dataset of 82 cases of intracranial neoplasms (679 individual measurements) that is processed with Machine Learning (ML) methods. As I understand, the authors see their main contributions (lines 353-355) in demonstrating that FFPE tissue samples are both suitable for vibrational spectroscopic examinations and that rapid spectroscopical examination and successive determination of biological tumor origin remains feasible.

In my opinion, the paper appears to be solid in its domain-related aspects (spectroscopy, brain tumors, etc.), but is somehow lacking in its machine learning-related part. The most severe disadvantage (which I think still corresponds to Minor revisions) is in the evaluation and comparison. Particularly, I believe that the authors need to add the comparison of the performance metrics obtained in their study to up-to-date baselines, which is the de-facto standard in ML publications.

For instance, in the Discussion the authors write (line 300): "in line with previous studies examining tumor necroses by means of optical spectroscopy [7,8,20,28]" - the authors need to be more specific regarding the studies mentioned. What exactly was "in line", and how do the ML model's performance metrics compare to the analogues (or to humans' performance) in the other studies?
How does their approach match the other diagnosis methods for tumor classification?
Since the dataset is not shared, it would be problematic for a reader to make her own comparison.

Another (related) issue is that the references in the paper contain no publications newer than 2021. Two years is a lot of time in Machine Learning, so it is recommended that the authors update the references and pick up-to-date baselines.
The above should aid the authors to more clearly justify the novelty of their study, which is currently not explicitly explained.

Methodology-wise, the paper is reasonably detailed, but some issues need clarification. Table 1 contains 9 tumors, but the authors work with 7 classes (cf. Fig. 3). I am not exactly an expert in tumors, so this transition might be obvious for a specialist in the field. Still, I would recommend the authors to explain how they arrive at their classification.

There is some non-conventional text in the References: "abgerufen am 13. November 2020".

Author Response

Point-by-Point response to reviewer 1

Manuscript-ID: brainsci-2892164

Title: Machine-Learning Assisted Classification of Paraffin-Embedded Brain Tumors Using Raman Spectroscopy

Corresponding author: Dr. Gilbert Georg Klamminger and Dr. Felix Kleine Borgmann; gil_bert@hotmail.de

We thank the reviewer for the positive assessment of our work and all the thoughtful comments that helped us to improve our manuscript. In the following, we will present how we addressed the requirements of the reviewer and incorporated all points raised into our manuscript.

All line numbers mentioned below refer to the updated and revised version of the manuscript.

Machine-Learning Assisted Classification of Paraffin-Embedded Brain Tumors Using Raman Spectroscopy

* For some reason, the paper's title in the PDF file does not match the one in the reviewing system. I assume that the authors might have updated their submission at some point and that the editors did not mind that.

As the reviewer already assumed, the manuscript title was adapted in full accordance with the Editorial staff. Unfortunately, the modification could not be initially transferred to the submission system and is here therefore not yet adapted.

The considered manuscript assesses the application of Raman Spectroscopy (RS) to formalin fixed paraffin-embedded (FFPE) intracranial neoplasms for classification of tumors into 7 classes. In this, the authors rely on the self-collected dataset of 82 cases of intracranial neoplasms (679 individual measurements) that is processed with Machine Learning (ML) methods. As I understand, the authors see their main contributions (lines 353-355) in demonstrating that FFPE tissue samples are both suitable for vibrational spectroscopic examinations and that rapid spectroscopical examination and successive determination of biological tumor origin remains feasible.

In my opinion, the paper appears to be solid in its domain-related aspects (spectroscopy, brain tumors, etc.), but is somehow lacking in its machine learning-related part. The most severe disadvantage (which I think still corresponds to Minor revisions) is in the evaluation and comparison. Particularly, I believe that the authors need to add the comparison of the performance metrics obtained in their study to up-to-date baselines, which is the de-facto standard in ML publications.

For instance, in the Discussion the authors write (line 300): "in line with previous studies examining tumor necroses by means of optical spectroscopy [7,8,20,28]" - the authors need to be more specific regarding the studies mentioned. What exactly was "in line", and how do the ML model's performance metrics compare to the analogues (or to humans' performance) in the other studies? How does their approach match the other diagnosis methods for tumor classification?

Since the dataset is not shared, it would be problematic for a reader to make her own comparison.

As the reviewer stated, vibrational spectroscopy on FFPE tissue is indeed challenging due various reasons (among others: degradation/fragmentation of nucleic acids, protein cross-linking, spectral signal of paraffin wax, modification of biological lipids related to the dewaxing process, see lines 142-146) compared to unprocessed tissue. Within our manuscript we therefore outline the general suitability of FFPE specimens for spectroscopic diagnostic in dependence of different clinically relevant use cases.

We thank the reviewer for the positively outlined aspects of our manuscript - in order to improve the mentioned ‘comparison to reported up-to-date results’, the following extensive additions have been implemented within our manuscript:

- General information about gold standards of brain tumor diagnosis and recent spectroscopic studies and their performance, see lines 120-123 and lines 131-138.

- published background information on the spectroscopic features of tumor necrosis and its relation to our results, see lines 402-411.

- information about spectroscopic tumor detection in comparison to imaging guided tumor detection, see lines 420-424.

- comparison and background information of spectroscopic performances in comparable research settings, see lines 428-434 and lines 439-444.

Another (related) issue is that the references in the paper contain no publications newer than 2021. Two years is a lot of time in Machine Learning, so it is recommended that the authors update the references and pick up-to-date baselines.

The above should aid the authors to more clearly justify the novelty of their study, which is currently not explicitly explained.

We thank the reviewer for the hint and added additional up-to-date references, for example: Zhang et al. (2023), Romanishkin et al. (2022), Quesnel et al. (2023), Jabarkheel et al. (2022), Klein et al. (2024), Mirizzi et al. (2024).

We do agree: together with the corresponding additional new background information added within the main text of the manuscript, as well as the re-structured “Conclusion” paragraph, we believe that our results presented appeal to the reader in a more comprehensive way.

Methodology-wise, the paper is reasonably detailed, but some issues need clarification. Table 1 contains 9 tumors, but the authors work with 7 classes (cf. Fig. 3). I am not exactly an expert in tumors, so this transition might be obvious for a specialist in the field. Still, I would recommend the authors to explain how they arrive at their classification.

We thank the reviewer for the comment.

Table 1 depicts all tumor measurements according to the underlying tumor entity.

Within the confusion matrix in Figure 3D, solely the primary brain tumors and brain metastasis are classified (the meningioma tumor group is not).

We rephrased the corresponding passage within our manuscript to avoid any potential confusion, see line 306-307.

There is some non-conventional text in the References: "abgerufen am 13. November 2020".

We deleted the non-conventional phrase.

Reviewer 2 Report

Comments and Suggestions for Authors

Dear Authors,

-> the abstract should provide more detailed information about the methodology used in the study. And the authors should include more keywords related to the topic.

 ->In the introduction, there is a lack of introductory information about the subject of brain tumors and Raman spectroscopy. 

->The methodology section is well-described, providing a clear outline of the experimental procedures.  

->The results section should have a final sentence of the main results.

->In the discussion section, the authors should additional data from relevant literature

->Furthermore, add a chapter of limitations 

 ->Finally, add a chapter on conclusions should be added to summarize the key findings 

->Additionally, the article could benefit from more references 

Overall, the article presents valuable research on the classification of brain tumors using Raman spectroscopy. With these suggested improvements, the impact and readability of the study will be greatly enhanced.

Author Response

Point-by-Point response to reviewer 2

Manuscript-ID: brainsci-2892164

Title: Machine-Learning Assisted Classification of Paraffin-Embedded Brain Tumors Using Raman Spectroscopy

Corresponding author: Dr. Gilbert Georg Klamminger and Dr. Felix Kleine Borgmann; gil_bert@hotmail.de

Overall, the article presents valuable research on the classification of brain tumors using Raman spectroscopy. With these suggested improvements, the impact and readability of the study will be greatly enhanced.

We thank the reviewer for the thoughtful comments on our manuscript and the positive feedback. In the following we address the points individually.

All line numbers mentioned below refer to the updated and revised version of the manuscript.

> the abstract should provide more detailed information about the methodology used in the study. And the authors should include more keywords related to the topic.

We rephrased the abstract with a special focus more detailed description of the methodology, such as CaF2 slides, random forest analysis as well as splitting into training and external validation cohorts, see lines 33, 35, 37).

We added more keywords, see lines 48, 49.

->In the introduction, there is a lack of introductory information about the subject of brain tumors and Raman spectroscopy.

We added additional background information about the subject of brain tumors and RS with a special focus on recently published literature, see lines 132-138.

Additionally, we added introductory general information about the current gold standard of brain tumor diagnostics, see lines 120-123.

->The methodology section is well-described, providing a clear outline of the experimental procedures. 

We thank the reviewer for the positive feedback.

->The results section should have a final sentence of the main results.

We added a final sentence to the result section, see lines 384-387.

->In the discussion section, the authors should additional data from relevant literature.

We thank the reviewer for the enriching comment and added published background information on the spectroscopic features of tumor necrosis and its relation to our results, see lines 402-411 as well as information about spectroscopic tumor detection in comparison to imaging guided tumor detection, see lines 420-424 and a comparison / more background information of spectroscopic performances in comparable research settings, see lines 428-434 and lines 439-444.

->Furthermore, add a chapter of limitations

We thank the reviewer for the hint; in fact, limitations are broadly discussed in line 445-483.

Nevertheless, we do agree that this should be clearly visible for the reader and changed the phrasing in line 447 accordingly as well as the formatting of line 448 (new paragraph as reflection).

->Finally, add a chapter on conclusions should be added to summarize the key findings

We adapted and rearranged the discussion section in order to have a conclusion section where findings are summarized, see lines 484-497.

->Additionally, the article could benefit from more references

We thank the reviewer for the hint and added additional up-to-date references, for example: Zhang et al. (2023), Romanishkin et al. (2022), Quesnel et al. (2023), Jabarkheel et al. (2022), Klein et al. (2024), Mirizzi et al. (2024).